# The Association of Problematic Internet Shopping with Dissociation among South Korean Internet Users

**DOI:** 10.3390/ijerph17093235

**Published:** 2020-05-06

**Authors:** Young-Mi Ko, Sungwon Roh, Tae Kyung Lee

**Affiliations:** 1Mental Health Research Institute, National Center for Mental Health, Seoul 04933, Korea; daquin07@korea.kr; 2Department of Psychiatry, Hanyang University, College of Medicine, Seoul 04763, Korea; swroh@hanyang.ac.kr; 3Department of Mental Health, ChunCheon National Hospital, ChunCheon City 24409, Korea

**Keywords:** compulsive buying, addictive shopping, impulsivity, online shopping, dissociation

## Abstract

*Background*: This study examined patterns of problematic shopping behavior by South Korean internet users to investigate the association between problematic internet shopping (PIS) and dissociative experiences.; *Methods*: Five hundred and ninety eight participants from 20–69 years old were recruited through an online panel survey. We gathered information about sociodemographic characteristics, alcohol use, caffeine intake, and online shopping behaviors. Psychopathological assessments included Korean version of dissociative experience scale (DES-K), Canadian Problem Gambling Index (CPGI-K), the modified Stress Response Inventory (SRI-MF), the Barratt Impulsive Scale-11-Revised (BIS-K). We used multiple logistic regression analysis with the Richmond compulsive buying scale (RCBS-K) as the dependent variable.; *Results*: The prevalence of shoppers with internet-based problem shopping was 12.5%. The amount of time spent on online shopping was correlated with PIS severity (OR = 1.008, *p* < 0.01). The risk of PIS was related to an increased tendency toward dissociation (OR = 1.044, *p* < 0.001) and impulsivity (OR = 1.046, *p* < 0.05). *Conclusions*: PIS participants with dissociation showed higher levels of perceived stress, gambling problems, and impulsivity than did PIS participants without dissociation. This study suggests that dissociation was associated with a higher burden of PIS as it was connected to poor mental health problems.

## 1. Introduction

With the spread of the internet, online shopping is also showing steady growth. The convenience offered by the internet contributes to its popularity. The internet provides a huge diversity of shopping information and simultaneous access to many online stores, thereby living up to expectations for immediate rewards and emotional enhancement.

South Korea is one of the best-connected countries in the world. In 2018, the internet penetration rate was at around 96 percent, while the internet usage rate was at 91.6 percent [1]. The South Korean government reported that online or internet-based shopping transaction value was 9.926 billion USD in July 2019, rising by 15.4% from July 2018 [2]. Because of an increase in high-speed internet access connections, lower connection costs, and increasing consumer competence, e-commerce activity has been rising.

For most people, buying is a normal and routine part of everyday life. Today, shopping is considered both a utilitarian and social or leisure activity with hedonistic features [3]. However, excessive shopping leads to distress or impairment. Compulsive and excessive spending is controversial, and such consumption and buying behaviors have been researched in both the business and medical literature.

The phenomenon of compulsive buying or an uncontrolled shopping spree has been described for 100 years in the literature [4,5]. Still, the phenomenon has not been included as a formal mental health disorder in the most recent, 5th edition of the Diagnostic and Statistical Manual of Mental Disorders (DSM-5) and the recently released 11th revision of the International Classification of Diseases (ICD-11) [6,7,8]. Uncontrolled buying behavior has been referred to in the literature as uncontrolled buying [9], compulsive buying [10], compulsive shopping [11], addictive buying [12], excessive buying [13], and pathological buying [14].

Given the rapid growth of online shopping, researchers have concerns about issues surrounding repetitive problematic behaviors associated with internet-based shopping. According to Rose and Dhandayudham [15], addictive online shopping may negatively influence not only an individual’s daily and social life but also their economic status. Zhao et al. [16] described a tendency to engage in excessive, compulsive, and problematic shopping behavior via the internet that results in consequences associated with economic, social, and emotional problems such as online shopping addiction. Müller et al. use the term “online buying-shopping disorder” [17]. In the current paper, we use “Problematic Internet Shopping (PIS)”, a more neutral expression in accordance with Lam and Lam [18].

Investigating the clinical characteristics of PISis of great importance as it will provide insights for developing diagnostic criteria. Improvement of understanding for PIS will enhance our capacity to recognize and define their presence. However, PIS is a complex and highly debated concept. Previous studies suggest that compulsive or pathological forms of buying behavior fit well into the behavioral addiction spectrum [19,20,21]. Disordered gambling, the only behavioral addiction being included in the main section of DSM-5 [6], is associated with problematic shopping [22,23]. Problematic shopping is associated with comorbid psychiatric conditions, such as mood disorders, and other behavioral addictions [14,24]. 

Behavioral addictions have been hypothesized as having similarities to substance addictions. Substance use disorders (SUD) co-occur with behavioral addictions. Higher caffeine intake has been associated with impulsivity and gambling [25]. In the case of Compulsive Buying, the co-morbidity rate with SUD was estimated as 21–46% [26]. Maraz et al. reported that Compulsive Buyers showed high impulsivity, high levels of distress, and substance use (alcohol, smoking, and illicit drug) in their shopping mall visitor sample [27]. These findings suggest that behavioral addictions share a common pathophysiology with SUD.

Some research has noted a correlation between behavioral addiction and dissociation [20,21]. Berancy and colleagues (2013) suggested that addiction is an absorbent relation established with an object that determines a person’s cognitive, emotional, and behavioral states, causing significant damage in different areas of life [28]. 

Ludwig [29] said, “Dissociation represents a process whereby certain mental functions which are ordinarily integrated with other functions presumably operate in a more compartmentalized or automatic way, usually outside the sphere of conscious awareness or memory recall.”

Maldonado and Spiegel [30] and DSM-5 [6] defines dissociation as the “disruption of and/or discontinuity in the normal integration of consciousness, memory, identity, emotion, perception, body representation, motor control, and behavior. Dissociative symptoms in mental disorders are of high clinical relevance”. 

Researchers have reported that symptoms of dissociation have been linked to maladaptive functioning, symptom severity, and poor treatment response in substance-related and addictive disorders [30,31].

The importance of dissociation in the psychopathology of addiction has been confirmed [32,33,34]. Somer [35] found that levels of dissociation negatively contributed to the prediction of abstinence. Craparo et al. [32] reported that addictive behaviors have a dissociative nature that allows individuals to manage negative and unregulated emotions. They suggested that dissociation is a predictor of addiction. 

Thus, the present exploratory study examined clinical characteristics and psychological aspects of PIS among South Korean Internet users. Based on the extant literature, we expected that PIS would be associated with elevated impulsivity, emotional distress, depression, more severe gambling problems, and substance abuse (alcohol and caffeine). Secondly, we investigate the relationship between PIS and dissociative symptoms. We hypothesized that dissociation would be positively associated with higher PIS severity and mental health problems. 

## 2. Methods

### 2.1. Participants

The total number of participants was 598, and we applied random sampling through proportionate allocation in accordance with the sex and age distribution of South Korea. Participants from 20–69 years were recruited through an online research service, ZINNOS R&C (ZINNOS R&C Co. Ltd., Seoul, Korea). The cross-sectional study was performed from August to September 2015. As this study aimed to collect research materials from dedicated, internet-based shoppers, this study was done through an online panel survey conducted by ZINNOS R&C, which, at the time of the survey, had 75,000 or more members.

Email invitations to complete an online survey were sent out to a random sample of potential respondents in the ZINNOS R&C panel. Of the 8977 Korean internet users who received an invitation, 598 (6.66%) visited the survey webpage. Participants completed a self-administered questionnaire that gathered information about sociodemographic characteristics, alcohol and caffeine use, online shopping behaviors, problematic buying, dissociation experiences, gambling problems, depression, stress perception, and impulsivity. Problematic shopping was measured with the Korean version of the Richmond Compulsive Buying Scale (RCBS). This study was approved by the National Center for Mental Health Institutional Review Board (NCMH 2015-06) and adhered to ethical policies.

### 2.2. Measures

#### 2.2.1. Sociodemographic Characteristics

We examined the subjects’ sex, age, and marital status. Participants aged 20–69 years were recruited, and for analysis were divided into age groups by decade: 20–29, 30–39, 40–49, 50–59, and 60–69. Marital status was divided into three groups: married, separated/widowed/divorced, and single.

#### 2.2.2. Internet Shopping Behaviors

The total time that participants had spent internet shopping, the amount of money they had spent on internet shopping in the past month, the time spent on activities related to shopping during the day, and the average number of days per week they engaged in activities related to shopping in the past month were surveyed and measured as continuous variables. Additionally, participants were asked whether they had the experience of making an online purchase in excess of their income. 

#### 2.2.3. Alcohol and Caffeine Use

Alcohol use was assessed with the question: “How many glasses do you drink?” For the question, “How many times have you drunk four or more glasses of an alcoholic beverage in the past year?” answers of “not at all,” “once a month,” “twice a month,” “once a week,” and “two or three times a week” were given. The question, “How many caffeine drinks (including coffee) do you drink a day on average?” was asked, and the answers were measured as a continuous variable. 

#### 2.2.4. The Korean Version of the Richmond Compulsive Buying Scale (RCBS-K)

The scale was developed by Ridgway et al. [36] and was used to measure the severity of PIS in this study. The scale consists of six questions that use a seven-point Likert scale. A score of 25 points or higher indicates that a respondent’s online shopping is a source of significant problems [36]. The RCBS has been reported to have high internal consistency, with a Cronbach’s alpha coefficient of 0.84, and evidence of validity [36]. For this study, the Cronbach’s alpha coefficient of RCBS-K was 0.906.

#### 2.2.5. The Korean Version of the Dissociative Experiences Scale (DES-K)

The Dissociative Experiences Scale (DES) is a 28-item self-report instrument. The DES was developed by Bernstein and Putman [37] and has adequate test-retest reliability, good split-half reliability, and good clinical validity. It can be completed in 10 min and scored in less than 5 min. It is easy to understand, and the questions are framed in a normative way that does not stigmatize the respondent for positive responses. The respondent clicks along a line anchored at 0% on the left and 100% on the right to show how often they have this experience. The overall DES score is obtained by adding up the answers of the 28 items and then dividing by 28; this yields an overall score ranging from 0 to 100. Scores higher than 20 on the Korean version of the DES (DES-K) may indicate the presence of a dissociative disorder. The DES-K has good validity and reliability, and overall good psychometric properties, exhibiting a Cronbach’s alpha coefficient of 0.94 [38]. The Cronbach’s alpha of DES-K for this study was 0.985.

#### 2.2.6. The Korean Version of the Canadian Problem Gambling Index (CPGI-K)

This scale was developed by Ferris and Wynne [39] to measure participant degree of gambling addiction. Scores for a total of nine questions are measured on a 4-point Likert-type scale (0: never, 1: sometimes, 2: frequently, 3: always). The range of the total score is from 0–27. According to the total score, the degree of gambling addiction is divided into nonproblematic gambling (0 points), low-danger gambling (1–2 points), mid-danger gambling (3–7 points), and problematic gambling (8 points or higher). The Korean version of the scale was standardized by Kim et al. [40], and the Cronbach’s alpha coefficient was 0.94. The Cronbach’s alpha of CPGI-K for this study was 0.947.

#### 2.2.7. The Korean Version of the Zung Self-Rating Depression Scale (ZDS-K)

The ZDS is a 20-item self-report measure of the symptoms associated with depression. Subjects rate each item with regard to how they have felt during the preceding week using a four-point Likert scale, with 4 representing the most unfavorable response. The sum of the 20 items, after transposing the 10 items that are reverse-scored, produces a raw score from 20–80. Previous studies have pointed out that scores are not meant to offer strict diagnostic criteria but rather denote levels of depressive symptoms that might be clinically significant [41,42]. The Korean version of the ZDS (ZDS-K) was used in this study and has high internal consistency (i.e., Cronbach’s alpha for the SDS = 0.79) [43]. The Cronbach’s alpha coefficient for ZDS-K in this study was 0.876

#### 2.2.8. The Modified Form of the Stress Response Inventory (SRI-MF)

SRI-MF is the short form of the Stress Response Inventory (SRI) was developed by Choi and colleagues [44] to score mental and physical symptoms occurring during the previous two weeks that might influence the current status of mental stress levels [45]. SRI scores can be categorized into seven stress factors: tension, aggression, somatization, anger, depression, fatigue, and frustration. The SRI assesses stress severity based on the stress symptoms or the effects of stressors. The SRI consists of 39 items that focus on emotional, somatic, cognitive, and behavioral stress responses. The SRI-MF consists of 22 items employing three factors: somatization, depression, and anger. The SRI-MF has good validity and reliability, exhibiting a Cronbach’s alpha coefficient of 0.93 [44]. The Cronbach’s alpha coefficient for SRI-MF in this study was 0.958.

#### 2.2.9. The Barratt Impulsive Scale-11-Revised (BIS-K).

The Barratt Impulsive Scale was developed to evaluate the degree of impulsiveness [46]. Sora Lee et al. [47] conducted a study on the reliability and validity of the Korean version of the scale. The scale has a total of 30 questions that are scored based on a four-point Likert scale. Sora Lee et al.’s (2012) study analyzed three sub-factors (cognitive impulsiveness, motor impulsiveness, and unplanned impulsiveness). Cronbach’s alpha for all questions was 0.78 (for cognitive impulsiveness: 0.623, for motor impulsiveness: 0.626, for unplanned impulsiveness: 0.580) [47]. For this study, the Cronbach’s alpha coefficient was 0.855.

### 2.3. Data Analysis

Investigators classified the participants into two groups (i.e., PIS and NPIS) based on their RCBS score. Byeon et al. [48] suggested that a score of 25 points or higher on the Korean version of the RCBS indicates a problematic shopper. We classified those whose RCBS score was 25 or more as the PIS group. Those whose RCBS-K score was below 25 were assigned to the non-PIS group (NPIS).

Chi-square tests of independence were employed to analyze the sociodemographic differences between the PIS and NPIS groups. For continuous variables, we used two-tailed t-tests to compare group mean differences. Mann–Whitney U test was adapted for non-normal distributed data. Pearson correlation analysis was used for data analysis. A multiple logistic regression analysis was used for four models. We estimated odds ratios (OR), adjusting for sex, age, and marital status (Model 1); adjusting for online shopping duration, online shopping amount, online shopping time, online shopping days, and experience of buying in excess of income (Model 2); adjusting for drinking and caffeine (Model 3); and adjusting for DES-K, CPGI-K, ZDS-K, SRI-MF, BIS-K (Model 4).

Model 4 is a full model. We considered statistical tests to be significant at an alpha level of 0.05 using a two-tailed test. We performed our data analyses using IBM SPSS Statistics version 21.0 (SPSS Inc., Chicago, IL, USA). 

## 3. Results 

### Descriptive Statistics

The sample size of the online panel survey was 598 adults. Their sociodemographic characteristics included 50.7% (*n* = 303) men and 49.3% (*n* = 295) women. Seventyfive (12.5%) participants were classified as PIS. Thirty-four (45.3%) men and 41 (54.7%) women scored 25 or above, with no difference between them. 

By age group, there was no statistically significant difference between PIS and non-PIS (NPIS) participants. As for marital status, the PIS participants were more often married than NPIS participants (Table 1). The PIS participants spent more money on shopping than did NPIS participants in the previous month. The PIS participants reported that they bought things more often in excess of their income than did NPIS participants.

With respect to shopping behaviors, PIS participants spent more money on shopping, more of their time on shopping-related activities, and had more buying experiences in excess of their income than did NPIS participants (Table 2). 

Compared to the NPIS participants, the PIS participants reported higher alcohol use. The PIS group also had higher scores on measures of dissociation, gambling severity, depression, perceived stress, and impulsivity (Table 3). Pearson’s correlation analysis revealed that the RCBS-K scores were positively related to DES-K, CPGI-K, ZDS-K, SRI-MF, and BIS-K (*p* < 0.01) (Table 4).

Table 5 presents the result of the multivariate logistic regression analysis. The dependent variable was the RCBS-K, and the reference group was NPIS. The OR was calculated from Model 1 with explanatory variables of sex, age, and marital status that are sociodemographic aspects, but no variable was statistically significant. The variable of online shopping behaviors was added to Model 2, while the variables of Model 1 were controlled. Among them, the OR value for the time spent on shopping-related activities was 1.010, showing a tendency toward an increased risk of PIS following an increase in the time spent on shopping-related activities (*p* < 0.001). Those reporting the experience of buying in excess of their income showed a tendency toward a greater risk for PIS than those who did not report such an experience (OR = 2.961, *p* < 0.001). Model 3 evaluated the impact of alcohol and caffeine consumption. The risk of PIS rose by frequency of alcohol consumption. Notably, those who consumed alcohol two or three times a week showed a higher OR of 2.88 than those who did not consume alcohol at all, and the value was statistically significant (*p* < 0.05). However, there is not significant relation for the amount of caffeine consumption with PIS.

Model 3 showed statistical significance between the time spent on shopping-related activities during a day (OR = 1.010, *p*< 0.001) and the experience of buying in excess of income (OR = 2.860, *p* < 0.001), both of which were still variables that were associated with PIS. Model 4 was a full model that assessed the impact of DES-K, CPGI-K, ZDS-K, SRI-MF, and BIS-K while all the variables of Model 3 were controlled. The result showed that the risk of PIS increased when the tendency toward pathological dissociation was higher (OR = 1.044, *p* < 0.001) and that the risk of PIS was higher when the level of impulsiveness was higher (OR = 1.046, *p* < 0.05). In addition, time spent on shopping-related activities was correlated with PIS severity (OR = 1.008, *p* < 0.01), as well as with the duration of online shopping, indicating that the higher the online shopping duration was, the higher the risk of PIS (OR = 1.093, *p* < 0.05).

When we compared mental health problems between PIS participants with dissociation and without dissociation, PIS participants with dissociation showed higher levels of perceived stress, gambling problems, and impulsivity than did PIS participants without dissociation (Table 6).

## 4. Discussion

In line with the rapid increase in e-commerce activities, there have been growing concerns about PIS. However, little is known about the clinical and psychopathological aspects of PIS. In this cross-sectional study, we found out a significant prevalence rate (12.5%) of PIS among South Korean internet users. The number was lower than in those reported in previous studies (17.7% [49] and 33.6% [17]). This rate difference might be due to the use of a different survey method: The participants in the Kukar-Kinney et al. [49] study were women. Muller et al. [17] analyzed the pooled data of treatment-seeking patients with shopping disorders. 

Contrary to previous findings which indicating that young people and women were more prone to manifest PIS [17,50], the results indicated no link between PIS and gender, age, and monthly income. Of the demographic variables, only marital status distinguished between the group, with PIS participants were more often single than NPIS members. The result is not consistent with past findings regarding link between online buying disorder and partnership status [17]. Some studies have argued that loneliness is an important reason why people are developing addictive behaviors. Andreassen et al. (2017) suggested that individuals who were no in a personal relationship were more prone to developing addictive social media use than people who had partners [51]. Elton-Marshall et al. (2018) presented that gambling to escape feeling of loneliness was linked with problem gambling severity. They suggested that being married was a protective factor against problem gambling severity [52].

The results showed that PIS participants connected to online shopping sites longer and more frequently, and they spent more money when online shopping. This result was consistent with previous studies [53,54]. Increasing the time spent using the internet has been considered an index of problematic use and possible addiction. Lemmens and Hendriks [55] indicated that the time spent playing online games was strongly related to internet addiction. 

Although the amount of coffee intake did not distinguish PIS from NPIS, the results showed that individuals with PIS had more mental health predicaments as they presented with an increase in depression, perceived stress, impulsiveness, gambling, alcohol use, and dissociative experiences than the NPIS group. This result provides support for PIS as a behavioral addiction requiring clinical recognition and treatment.

In the current study, PIS participants’ gambling severity was higher than NPIS participants. Although disordered gambling is the only behavioral addiction classified in DSM-5, problematic gambling was categorized into the Impulse-Control Disorder section in DSM-IV. In this regard, the association of PIS with gambling indicates that PIS is a behavioral addiction. 

The relationship of PIS with impulsivity was consistent with previous studies. Impulsivity has been found to play an important role in the occurrence of addiction-related disorders [56,57]. Billieux et al. [58] reported that compulsive buying was positively correlated with impulsivity, and impulsivity was the most significant predictor of compulsive buying. Black [4] argued that pathological or compulsive buying should be classified as an impulse control disorder.

The key finding here was that the best-fit logistic regression model identified dissociation and impulsivity as being associated with PIS. The importance of dissociation in the psychopathology of addiction has been confirmed [32,33,34]. The literature indicates that symptoms of dissociation are present in a variety of mental disorders and connect to higher-burden illnesses and poorer treatment response [31,33,59,60,61]. 

As hypothesized, participants with PIS and dissociation exhibited poorer mental health, including higher stress levels, gambling, and impulsivity. According to Jacobs [34], dissociative symptoms resemble detachment states accompanying the acting-out phase of impulse control disorder. Maldonado and Spiegel (2019) indicated that dissociation represents more a disturbance in the organization or structure of mental contents than a disturbance in the mental contents themselves [30]. Lyssenko et al. (2018) suggested that the experience of dissociation can induce stress itself because it not only disrupts neurocognitive functioning but can also be perceived as losing control [31]. Kianpoor and Bakhshani (2012) reported that dissociation related with high-risk behaviors such as violence, heavy drinking, the use of illicit drugs, and dangerous driving [62]. From a clinical perspective, this finding underlines the importance of careful evaluation of dissociation symptoms. It will help health professionals to have recognition of people prone to high-risk behaviors as well as implement more effective strategies to prevent high-risk behaviors among at-risk populations. 

The current research has important implications for the prevention of problematic internet shopping. As the results were reviewed, our data pointed to the importance of interventions aimed at helping internet-using shoppers increase their level of awareness to prevent dissociation. For example, introducing a time or monetary limit pop-up reminder into the internet-based shopping mall sites might help control impulsiveness. Stewart and Wohl [63] suggested that a monetary limit pop-up reminder was effective for internet gamblers to facilitate adherence to monetary limits.

### Limitations

Despite this being one of the first studies to explore the clinical and psychopathological characteristics of PIS, this study has several limitations that might influence the generalizability of the findings. First, the use of cross-sectional, self-reported data in this study might have influenced our results through common method bias (e.g., causality problems; a prevalence-incidence bias) [64]. Second, because we used self-report measures instead of structured interviews, we cannot make a clinical diagnosis of dissociative disorders or clinically assess the relationship of PIS with different types of dissociative disorders (e.g., dissociative identity disorder, dissociative amnesia, depersonalization/derealization disorder). Third, we did not screen participants for the presence of traumatic psychological experiences, which can play an important role as the cause of dissociative disorders. Hence, we could not evaluate whether the dissociative disorder is a consequence of psychological trauma-related dissociative disorders. Finally, it is worth noting that we recruited participants through an online research service, ZINNOS R&C, which operates its own independent consumer panel in which participants are preregistered. The sample might not be representative of South Korean internet-based online shoppers or the general public. Besides, the response rate of this online panel study was 6.6%. The low rate may reflect the fact that only the panel members who have interested in the topic may have answered the invitation. Callegaro et al. [65] indicated that completion rates of online panel studies had a large variability going from 3%–91% with an average of 18%.

## 5. Conclusions

This study suggests that increasing awareness to prevent dissociation is important for addressing PIS. Individuals with PIS and dissociation showed a more severe mental health status than the PIS participants without dissociation. PIS participants with dissociation showed higher levels of perceived stress, gambling problems, and impulsivity than did PIS participants without dissociation. Introducing measures to prevent PIS and triggering of the state of detachment might increase a person’s ability to tolerate negative affect.

## Figures and Tables

**Table 1 ijerph-17-03235-t001:** Comparison of online shopper with and without problematic shopping behavior.

Variable		Demographics			Group Difference	
		PIS *n* = 75 (%), Mean (± SD)	NPIS *n* = 523 (%), Mean (± SD)	Total *n* = 598 (%), Mean (± SD)	χ2 /t	*p* Value
Gender	Male	34 (45.3)	269 (51.4)	303 (50.7)	0.977	0.323
	Female	41 (54.7)	254 (48.6)	295 (49.3)		
Age (years)	20–29	18 (24.0)	87 (16.6)	105 (17.6)	3.090	0.543
	30–39	16 (21.3)	114 (21.8)	130 (21.7)		
	40–49	17 (22.7)	131 (25.0)	148 (24.7)		
	50–59	17 (22.7)	120 (22.9)	137 (22.9)		
	60–69	7 (9.3)	71 (13.6)	78 (13.0)		
Marital status	Married	41 (54.7)	329 (62.9)	370 (61.9)	6.476	0.039
	Separated/ widowed/divorced	2 (2.7)	39 (7.5)	41 (6.9)		
	single	32 (42.7)	155 (29.6)	187 (31.3)		

PIS, Problematic Internet Shopping; NPIS, Non-Problematic Internet Shopping.

**Table 2 ijerph-17-03235-t002:** Comparisons of online shopping pattern between PIS and NPIS participants.

		Shopping Pattern			Group Differences	
		PIS	NPIS	Total	χ2 /t	*p* Value
		Mean or *n* (*SD or %*)	Mean or *n* (*SD or %*)	Mean or *n* (SD or %)		
Online shopping duration	year	11.06 (5.19)	10.37 (4.56)	10.46 (4.64)	1.202	0.230
Online shopping amount of budget	10k Won	46.69 (56.30)	27.84 (44.96)	30.21 (46.90)	2.776	0.007
Time spent shopping per day	minutes	109.84 (91.10)	47.26 (55.49)	55.12 (64.43)	5.796	0.000
The number of days shopping per week	days	4.37 (1.92)	3.33 (1.87)	3.46 (1.91)	4.477	0.000
Buying in the excess over the income	No	37 (49.3)	427 (81.6)	464 (77.6)	39.386	0.000
	Yes	38 (50.7)	96 (18.4)	134 (22.4)		

PIS, Problematic Internet Shopping; NPIS, Non-Problematic Internet Shopping.

**Table 3 ijerph-17-03235-t003:** Mental health problems in participants with and without PIS.

Variables		Mental Health			Group Difference	
		PIS Mean or *n* (SD)	NPIS Mean or *n* (SD)	Total	χ2 /t	*p* Value
Alcohol use	Not at all	8 (10.7)	96 (18.4)	104 (17.4)	11.968	0.018
	Once a month	10 (13.3)	134 (25.6)	144 (24.1)		
	Twice a month	16 (21.3)	99 (18.9)	115 (19.2)		
	Once a week	17 (22.7)	88 (16.8)	105 (17.6)		
	Two or three times a week	24 (32.0)	106 (20.3)	130 (21.7)		
Caffeine use	cups	2.71 (1.84)	2.66 (2.01)	2.66 (1.98)	0.199	0.842
DES-K		41.78 (24.43)	14.30 (14.50)	17.75 (18.46)	9.502	0.000
CPGI-K		6.16 (6.79)	1.14 (2.96)	1.77 (4.02)	6.319	0.000
ZDS-K		50.47 (8.09)	42.33 (9.13)	43.35 (9.39)	8.017	0.000
SRI-MF		41.24 (17.09)	21.84 (15.57)	24.27 (17.02)	9.963	0.000
BIS-K		69.69 (10.30)	58.46 (9.14)	59.87 (10.00)	9.791	0.000

DES-K, The Korean version of the Dissociative Experiences Scale; CPGI-K, The Korean version of the Canadian Problem Gambling Index; ZDS-K, The Korean version of ZungSelf-Rating Depression Scale; SRI-MF, The modified form of the Stress Response Inventory; BIS-K, The Korean version of The Barratt Impulsive Scale-11-Revised.

**Table 4 ijerph-17-03235-t004:** Reliabilities and Correlation between RCBS-K and other psychological scales.

Psychological Scales	Cronbach’s Alpha	95% CI	r(*p*)
RCBS-K	0.906	0.894–0.917	1
DES-K	0.985	0.984–0.987	0.559 **
CPGI-K	0.947	0.941–0.954	0.422 **
ZDS-K	0.876	0.861–0.890	0.356 **
SRI-MF	0.958	0.953–0.963	0.461 **
BIS-K	0.855	0.837–0.871	0.484 **

** *p* < 0.01. RCBS-K, The Korean version of the Richmond Compulsive Buying Scale; DES-K, The Korean version of the Dissociative Experiences Scale; CPGI-K, The Korean version of the Canadian Problem Gambling Index; ZDS-K, The Korean version of Zung Self-Rating Depression Scale; SRI-MF, The modified form of the Stress Response Inventory; BIS-K, The Korean version of the Barratt Impulsive Scale-11-Revised.

**Table 5 ijerph-17-03235-t005:** Contributing factors to RCBS-K identified in the multiple logistic regression.

Variable	Model 1				Model 2				Model 3				Model 4			
	B	S.E.	OR	*p* Value	B	S.E.	OR	*p* Value	B	S.E.	OR	*p*-Value	B	S.E.	OR	*p* Value
Gender (ref:male)																
female	0.253	0.250	1.288	0.310	–0.168	0.285	0.845	0.555	0.111	0.304	1.118	0.714	0.090	0.387	1.094	0.817
Age (ref:60~69)																
20~29	0.138	0.580	1.148	0.813	0.092	0.655	1.096	0.889	–0.002	0.658	0.998	0.997	–0.539	0.844	0.584	0.523
30~39	0.027	0.512	1.028	0.958	–0.436	0.574	0.647	0.448	–0.694	0.582	0.499	0.233	–1.116	0.753	0.328	0.138
40~49	0.121	0.482	1.128	0.802	0.266	0.531	1.304	0.617	0.048	0.536	1.050	0.928	–0.253	0.676	0.777	0.709
50~59	0.333	0.476	1.395	0.484	0.385	0.527	1.470	0.465	0.295	0.525	1.343	0.574	0.093	0.668	1.098	0.889
Marital status(ref:married)																
seperated/widowed/divorced	–0.915	0.747	0.400	0.221	–1.098	0.795	0.333	0.167	–1.097	0.790	0.334	0.165	–1.245	0.892	0.288	0.163
single	0.552	0.361	1.737	0.126	0.478	0.385	1.613	0.214	0.628	0.396	1.874	0.113	0.980	0.499	2.664	0.050
Online shopping duration					0.039	0.034	1.039	0.253	0.054	0.035	1.055	0.119	0.089	0.040	1.093	0.026
Online shopping amount					0.003	0.002	1.003	0.177	0.002	0.002	1.002	0.336	0.000	0.003	1.000	0.964
Time spent shopping per day					0.010	0.002	1.010	0.000	0.010	0.002	1.010	0.000	0.008	0.002	1.008	0.001
The number of days shopping per week					0.099	0.079	1.104	0.210	0.101	0.080	1.106	0.211	0.180	0.097	1.197	0.063
Buying exceeding the income (ref:No)																
Yes					1.086	0.287	2.961	0.000	1.051	0.294	2.860	0.000	0.608	0.372	1.837	0.102
Alcohol use (ref:Not at all)																
Once a month									–0.293	0.539	0.746	0.586	–0.275	0.601	0.759	0.647
Twice a month									0.552	0.505	1.737	0.274	0.111	0.617	1.117	0.858
Once a week									0.972	0.505	2.643	0.054	0.614	0.636	1.847	0.335
Two or three times a week									1.058	0.488	2.880	0.030	1.017	0.579	2.765	0.079
Caffeine consumption									–0.042	0.073	0.959	0.569	–0.018	0.100	0.982	0.854
DES-K													0.043	0.010	1.044	0.000
CPGI-K													0.075	0.042	1.078	0.073
ZDS-K													0.006	0.029	1.006	0.829
SRI-MF													0.021	0.014	1.021	0.131
BIS-K													0.045	0.022	1.046	0.035
-2LL			442.969		371.731				359.010				247.218			

RCBS-K, The Korean version of the Richmond Compulsive Buying Scale; DES-K, The Korean version of the Dissociative Experiences Scale; CPGI-K, The Korean version of the Canadian Problem Gambling Index; ZDS-K, The Korean version of ZungSelf-Rating Depression Scale; SRI-MF, The modified form of the Stress Response Inventory; BIS-K, The Korean version of the Barratt Impulsive Scale-11-Revise.

**Table 6 ijerph-17-03235-t006:** Comparison of mental health problems in PIS participants with and without dissociation.

Variables	Mental Health		Group Difference	
	Normal (*n =* 19) Mean Rank	Dissociation (*n =* 56) Mean Rank	M-W U	*p* Value
CPGI-K	25.24	42.33	289.50	0.003
SRI-MF	25.42	42.27	293.00	0.004
BIS-K	26.18	42.01	307.50	0.006

PIS, Problematic Internet Shopping, CPGI-K, The Korean version of the Canadian Problem Gambling Index; SRI-MF, The modified form of the Stress Response Inventory; BIS-K, The Korean version of the Barratt Impulsive Scale-11-Revised; M-W U Mann–Whitney U.

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
