# Peer review of "The Association of Problematic Internet Shopping with Dissociation among South Korean Internet Users"

_ijerph, 2020, doi:10.3390/ijerph17093235_

Round 1

Reviewer 1 Report

Reviewer’s report

Title: The association of problematic internet shopping with dissociation among South Korean internet users

The authors' idea of investigating the association between problematic internet shopping (PIS) and dissociative symptoms is very interesting. The paper is well written and easy-to-follow. The review of previous findings, the methodological aspects, the statistical analysis and the clearness of the results are the main strong points of this work. In short, it is a well-built paper and I only have a few comments related to the discussion, and some inaccuracies, that I hope to be helpful in improving the manuscript.

NO major comments.

Minor comments:

  1.  Keywords. It is advisable to include the term "dissociation”.
  2. Discussion section

In general, the considerations in this section refer to the appropriateness of further reflection and/or explanation on some of the results. For example:

- Given that the result “of the demographic variables, only marital status distinguished between the group, with PIS participants more often single than NPIS members" (line 291) is not consistent with the one obtained in the previous literature, could the authors advance any tentative explanatory hypothesis in this regard?

- By comparing mental health problems in PIS participant with and without dissociation, it is obtained that “PIS participants with dissociation showed higher level of perceived stress, gambling problem and impulsivity than did PIS participants without dissociation”. Although the authors advance some ideas on line 318, we consider it necessary to go deeper into this result.

  1. References section:

Review this section as there are some inaccuracies. For example,

Complete references: 

  • Maldonado, J.R.; Spiegel, D. Dissociative disorders. 2008. Include complete reference: Maldonado, J. R., & Spiegel, D. (2008). Dissociative disorders. In R. E. Hales, S. C. Yudofsky, & G. O. Gabbard (Eds.), The American Psychiatric Publishing textbook of psychiatry (p. 665–710). American Psychiatric Publishing, Inc.
  • Schimmenti, A.; Guglielmucci, F.; Barbasio, C.P.; Granieri, A. Attachment disorganization and 493 dissociation in virtual worlds: A study on problematic Internet use among players of online 494 role playing games. 2012. Add: Clinical Neuropsychiatry, 9, 5, 195-202
  • Homogenize the name of the journals: use abbreviations or full name (e.g., Compr Psychiatry -in reference nº 17 vs Comprehensive Psychiatry -in reference nº 51-) and include capital letters in the initials of the journal’s name (e.g., The Journal clinical psychiatry, Journal of palliative medicine).
  • Eliminate capital letters in references 41 y 42.

Author Response

Comments and Suggestions for Authors

Reviewer’s report

Title: The association of problematic internet shopping with dissociation among South Korean internet users

The authors' idea of investigating the association between problematic internet shopping (PIS) and dissociative symptoms is very interesting. The paper is well written and easy-to-follow. The review of previous findings, the methodological aspects, the statistical analysis and the clearness of the results are the main strong points of this work. In short, it is a well-built paper and I only have a few comments related to the discussion, and some inaccuracies, that I hope to be helpful in improving the manuscript.

NO major comments.

Minor comments:

  1.  Keywords. It is advisable to include the term "dissociation”.

RESPONSE: Thank you for the suggestion. We have added “dissociation” to Keywords.

 2. Discussion section

In general, the considerations in this section refer to the appropriateness of further reflection and/or explanation on some of the results. For example:

- Given that the result “of the demographic variables, only marital status distinguished between the group, with PIS participants more often single than NPIS members" (line 291) is not consistent with the one obtained in the previous literature, could the authors advance any tentative explanatory hypothesis in this regard?

RESPONSE: Thank you for your insightful review. We have added some explanation and incited some study results back up our thoughts for that.

 Some studies have argued that loneliness is an important reason why people are developing addictive behaviors. Andreassen et al (2017) suggested that individuals who were no in a personal relationship were more prone to developing addictive social media use than people who had partners. Elton-Marshall et al. (2018) presented that gambling to escape feeling of loneliness was linked with problem gambling severity. They suggested that being married was a protective factor against problem gambling severity.

- By comparing mental health problems in PIS participant with and without dissociation, it is obtained that “PIS participants with dissociation showed higher level of perceived stress, gambling problem and impulsivity than did PIS participants without dissociation”. Although the authors advance some ideas on line 318, we consider it necessary to go deeper into this result.

RESPONSE: Thank you for your recommendation. Following your suggestion, we have added sentences to support the information.

 Maldonado and Spiegel (2019) indicated that dissociation represents more a disturbance in the organization or structure of mental contents than a disturbance in the mental contents themselves.

Lyssenko et al. (2018) suggested that the experience of dissociation can induce stress itself because it not only disrupts neurocognitive functioning but can also be perceived as losing control.

Kianpoor and Bakhshani (2012) reported that dissociation related with high-risk behaviors such as violence, heavy drinking, the use of illicit drugs, and dangerous driving.

From a clinical perspective, this finding underlines the importance of careful evaluation of dissociation symptoms. It will help health professionals to have recognition of people prone to high-risk behaviors as well as implement more effective strategies to prevent high-risk behaviors among at-risk populations.

3. References section:

Review this section as there are some inaccuracies. For example,

Complete references: 

  • Maldonado, J.R.; Spiegel, D. Dissociative disorders. 2008Include complete reference: Maldonado, J. R., & Spiegel, D. (2008). Dissociative disorders. In R. E. Hales, S. C. Yudofsky, & G. O. Gabbard (Eds.), The American Psychiatric Publishing textbook of psychiatry(p. 665–710). American Psychiatric Publishing, Inc.
  • Schimmenti, A.; Guglielmucci, F.; Barbasio, C.P.; Granieri, A. Attachment disorganization and 493 dissociation in virtual worlds: A study on problematic Internet use among players of online 494 role playing games. 2012Add: Clinical Neuropsychiatry, 9, 5, 195-202
  • Homogenize the name of the journals: use abbreviations or full name (e.g., Compr Psychiatry -in reference nº 17 vsComprehensive Psychiatry -in reference nº 51-) and include capital letters in the initials of the journal’s name (e.g., The Journal clinical psychiatry, Journal of palliative medicine).
  • Eliminate capital lettersin references 41 y 42.

RESPONSE: Thank you for your detailed review. We have corrected the references

Reviewer 2 Report

The article is a very good work. It has a very clear introduction, objectives and methodology are well described, results are concise and discussion has a very good relation with prior studies. 

Some minor rectifications should be considered:

  • Line 86-87. The definition of the DSM 5 is almost the same of Maldonado and Spiegel. I recommend to cite both together, rather repeat the definition: Maldonado and Spiegel (30) and DSM 5 (6) described dissociation as...
  • Lines 112-113. Add the ratio of response (6.66%). Some explication about why this ratio was so low it would be necessary. If they have information about other ratios in similar studies, they should write about it. 
  • Lines 132-133. Says"How many times have you drunk four or fewer glasses of an alcoholic beverage in the past year?”. I suppose that is "four o more glasses". Correction is necessary.
  • Lines 142-143. It sound very strange that alpha coefficients are exactly the same for the original study (36) and for this. I recommend to check it.
  • Line 193. Authors present only alpha coefficient of the total BIS-K, what are the alpha coefficients of the subscales for this study?
  • Line 207. After saying that model 4 is a full model, it would be better to give details about full model (results of scales were added: DES-K, CGPI-K, ZDS-K, SRI-MF, BIS-K)
  • Lines 259-260. Authors say: ...caffeine consumption was not correlated. There is not correlation in a regression model. It is better to say "there is not significant relation for the amount of caffeine consumption.
  • Lines 260-261. Signifiant results of model 3 are added, however this are almost the same than in model 3 (first version), so I consider that it is not necessary write down this information. Moreover, this information is in table 5.
  • Line 270. It is necessary to present footnote of table 5.
  • Line 276. The name of the variables of table 6 are in bad format.
  • Line 338. Authors do not present explanation about what could be the reasons of participants for responding to this study. Did they receive information about the objectives of the study? This information could influence people to accept or reject participation. Here, again, they should emphasize that tax of response was very low 6.66%

Author Response

The article is a very good work. It has a very clear introduction, objectives and methodology are well described, results are concise and discussion has a very good relation with prior studies. 

Some minor rectifications should be considered:

  • Line 86-87. The definition of the DSM 5 is almost the same of Maldonado and Spiegel. I recommend to cite both together, rather repeat the definition: Maldonado and Spiegel (30) and DSM 5 (6) described dissociation as...

RESPONSE: We appreciate your recommendation. We have revised the sentence following your suggestion.

Maldonado and Spiegel [30] and DSM-5 [6] defines dissociation as the “disruption of and/or discontinuity in the normal integration of consciousness, memory, identity, emotion, perception, body representation, motor control, and behavior. Dissociative symptoms in mental disorders are of high clinical relevance”.

  • Lines 112-113. Add the ratio of response (6.66%). Some explication about why this ratio was so low it would be necessary. If they have information about other ratios in similar studies, they should write about it. 

RESPONSE: Thank you for your detailed review. We have marked the ratio and added the information at limitation section.

Of the 8,977 Korean internet users who received an invitation, 598 (6.66%) visited the survey webpage.

Besides, the response rate of this online panel study was 6.66%. The low rate may reflect the fact that only the panel members who have interested in the topic may have answered the invitation. Callegaro et al.[66] indicated that completion rates of online panel studies had a large variability going from 3%-91% with an average of 18%.

  • Lines 132-133. Says"How many times have you drunk four or fewerglasses of an alcoholic beverage in the past year?”. I suppose that is "four o more glasses". Correction is necessary.

RESEPONSE: Thank you for your detailed review. We have corrected the typo.

  • Lines 142-143. It sound very strange that alpha coefficients are exactly the same for the original study (36) and for this. I recommend to check it.

RESPONSE: Thank you for your detailed review. We have revised the sentences.

The RCBS has been reported to have high internal consistency, with a Cronbach's alpha coefficient of 0.84, and evidence of validity [37]. For this study, the Cronbach’s alpha coefficient of RCBS-K was 0.906.

  • Line 193. Authors present only alpha coefficient of the total BIS-K, what are the alpha coefficients of the subscales for this study?

RESPONSE: Following your recommendation, we have presented the alpha coefficients for the subfactors.

For this study, the Cronbach’s alpha for all questions was 0.855 (for cognitive impulsiveness: 0.629, for motor impulsiveness: 0.790, for unplanned impulsiveness: 0.653)

  • Line 207. After saying that model 4 is a full model, it would be better to give details about full model (results of scales were added: DES-K, CGPI-K, ZDS-K, SRI-MF, BIS-K)

RESPONSE: Thank you for your suggestion. We have revised the sentence.

We estimated odds ratios (OR), adjusting for sex, age, and marital status (Model 1); adjusting for online shopping duration, online shopping amount, online shopping time, online shopping days, and experience of buying in excess of income (Model 2); adjusting for drinking and caffeine (Model 3); and adjusting for DES-K, CPGI-K, ZDS-K, SRI-MF, BIS-K (Model 4).

  • Lines 259-260. Authors say: ...caffeine consumption was not correlated. There is not correlation in a regression model. It is better to say "there is not significant relation for the amount of caffeine consumption.

RESPONSE: We really appreciated your recommendation. We have changed the expression following your suggestion.

  • However, there is not significant relation for the amount of caffeine consumption.

  • Lines 260-261. Signifiant results of model 3 are added, however this are almost the same than in model 3 (first version), so I consider that it is not necessary write down this information. Moreover, this information is in table 5.

RESPONSE: Thank you for your insightful review. Following your suggestion, We have deleted the sentence.

  • Line 270. It is necessary to present footnote of table 5.

RESPONSE: Thank you for the recommendation. We have presented footnote to Table 5.

RCBS-K, The Korean version of the Richmond Compulsive Buying Scale; DES-K, The Korean version of the Dissociative Experiences Scale; CPGI-K, The Korean version of the Canadian Problem Gambling Index; ZDS-K, The Korean version of Zung Self-rating Depression Scale; SRI-MF, The modified form of the Stress Response Inventory; BIS-K, The Korean version of the Barratt Impulsive Scale-11-Revised

Line 276. The name of the variables of table 6 are in bad format.

RESPONSE: Thank you for your detailed review. We have revised the name of variables in table 6.

Table 6. Comparison of mental health problems in PIS participants with and without dissociation

Variables

Mental Health

Group difference

Normal (n=19)

Mean Rank  

Dissociation (n=56)

Mean Rank

M-W U

p-value

CPGI-K

25.24

42.33

289.50

0.003

SRI-MF

25.42

42.27

293.00

0.004

BIS-K

26.18

42.01

307.50

0.006

PIS, Problematic Internet Shopping, CPGI-K, The Korean version of the Canadian Problem Gambling Index; SRI-MF, The modified form of the Stress Response Inventory; BIS-K, The Korean version of the Barratt Impulsive Scale-11-Revised; M-W U Mann-Whitney U

  • Line 338. Authors do not present explanation about what could be the reasons of participants for responding to this study. Did they receive information about the objectives of the study? This information could influence people to accept or reject participation. Here, again, they should emphasize that tax of response was very low 6.66%

RESPONSE: We acknowledge the reviewer’s opinion. The response rate of the online panel was 6.6%. The completion rate of online panel studies has been known to have a large variability. Callegaro and his associates pointed out that the rate of completion in online panel was going from 3%-91% with an average of 18%. We have added the weak point of this study at the limitation sentences.

  • Besides, the response rate of this online panel study was 6.6%. The low rate may reflect the fact that only the panel members who have interested in the topic may have answered the invitation. Callegaro et al. indicated that completion rates of online panel studies had a large variability going from 3%-91% with an average of 18%.

Callegaro, M.; Baker, R.P.; Bethlehem, J.; Göritz, A.S.; Krosnick, J.A.; Lavrakas, P.J. Online panel research: A data quality perspective; John Wiley & Sons: 2014.

This manuscript is a resubmission of an earlier submission. The following is a list of the peer review reports and author responses from that submission.

Round 1

Reviewer 1 Report

General Comments: Thanks for giving me the chance to review the manuscript “The association of problematic internet shopping with dissociation among South Korean internet users”. I think it is interesting. The topic is timely, and method and result sections are sound. Nevertheless, I have some further minor amendments. Overall, there are a few deficiencies in the paper. there are certain language problems although the meaning is generally clear. It is strongly recommended that the authors seek the assistance of someone well versed in English to help with the language. Specific comments: 1. In abstract, information such as questionnaires needs to be supplemented. 2. The literature review is incomplete. The current version does not adequately review and analyze the related literature involved in this research, which leads to unclear logical frameworks for research questions, rigorous research assumptions, and inability to experience the scientificity and advanced nature of this research. 3. The focus of this research is problematic Internet shopping. It can be seen from the method section that this study uses several questionnaires(RCBS-K、DES-K、 CPGI-K、ZDS-K、BIS-K), but the literature review section does not explain the relationship between the use of these questionnaires and problematic Internet shopping from the perspective of a theoretical framework. 4. The discussion section also needs to add some further information. It is suggested to add the discussion of the relationship between problematic internet shopping and stress, problem gambling,depression,impulsive, et al. The content of the discussion needs to based on a comparative analysis with existing research, focus on problematic Internet shopping, and then discuss related influencing factors.

Reviewer 2 Report

The manuscript investigates the association between problematic internet shopping (PIS) behavior and dissociative experiences. The methodology bases on regression analysis of behavior and clinical parameters obtaine through on-line survey in a population of almost six hundred subjects. The results put into evidence a correlation of PIS with time spent on-line, with tendency towards dissociation and with impulsivity.

Strengths of the research are the high number of subjects included in the analysis and the good reference to previous literature; on the other hand, there are some weaknesses that are listed below:

Lack of clinical figures in the survey administration. A major limitation of the study concerns the absence of a medical expert in support during the survey administration. The data gathered through self-reported online surveys could have seriously suffered from biases related to social desirability, given the investigated research topic. Some of the analysed constructs appear to be poorly supported. Constructs including problem gambling, depression rate, alcohol, and caffeine use result to be scarcely introduced in the opening literature part. A sheer theoretical grounding has to be provided for inclusion in the analysis of such constructs and the lack of parallel constructs (e.g. smoking habits, drug use, etc.). Their importance in the Korean setting has to be stated as well in order to foster the generalisability of the results. The comparison of mental health problems in PIS participants results to be scarcely supported by sample size. Specifically, the sample size of one of the two sub-groups (n=18) object of comparison appears to be too little. The conclusion that "individuals with PIS dysplayed increased online shopping duration and impulsiveness" seems not relevant as increased online shopping duration and impulsiveness can be the definition of PIS.

Reviewer 3 Report

The authors present an interesting study from South Korea on PIS. The following points should still be revised:
1. Why the consumption of alcohol and caffeine is included in the analyses is not yet explained in the introduction. This should still be done, otherwise these variables need not be included in the analyses.
2. On page 3, the text does not agree with the table. It reads: "As for marital status, the PIS participants were more often married than NPIS participants." However, according to the table, the PIS group is more often single.
3. All KRW figures in the text could be converted into US dollars to make the corresponding explanations understandable to an international audience.
4. A question about Table 4: Does the RCBS-K scale is used in its dichotomous or interval-scaled form?
5. It would be necessary to have a correlation table for all variables included in the text or appendix, which would help to understand the sometimes strong changes in the coefficients in Table 5 when other variables are taken into account.
6. Information on social status (education, unemployment, income, etc.) was not asked in the survey? These would also be important control variables.
7. How many respondents are ultimately included in the multivariate analysis? Were there missing values for individual variables and how were they dealt with if necessary?